# Booster immunization of meningococcal meningitis vaccine among children in Hangzhou, China, 2014-2019

**Xinren Che, Yan Liu\*, Jun Wang, Yuyang Xu, Xuechao Zhang, Wenwen Gu, Wei Jiang, Jian Du, Xiaoping Zhang**

Department of Expanded Program on Immunization, Hangzhou Center for Disease Control and Prevention, Hangzhou, Zhejiang, China

\* smileforever81@126.com

**Data Availability Statement:** All relevant data are within the paper and its Supporting information files.

## Abstract

### Background

Despite China's Expanded Program on Immunization (EPI) provides 2 doses of group A and group C meningococcal polysaccharide vaccine (MPV-AC) for children at 3 years and 6 years old, more self-paying group ACYW135 meningococcal polysaccharide vaccines (MPV-ACYW135) have been used as an alternative to MPV-AC to prevent Neisseria meningitidis serogroup C,Y,W135. We provide recommendations for Chinese booster immunization of meningococcal meningitis vaccine by analyzing the service status of MPV-AC and MPV-ACYW135.

### Methods

Reported data of routine immunization coverage from all districts of Hangzhou registered in the China Information Management System For Immunization Programming (CIMSFIP) between 2014 to 2019 were described and evaluated. Descriptive epidemiological methods were used to characterize the data. Adverse event following immunization (AEFI) were collected from Chinese national adverse event following immunization information system (CNAEFIIS) to compare the safety of MPV-AC and MPV-ACYW135.

### Results

1376919 doses of booster immunization of meningococcal meningitis vaccine (MenV) in CIMSFIP were conducted in China Hangzhou from 2014 to 2019, with reported immunization coverage rates above 95%. The proportion of children using MPV-ACYW135 increased from 12.63% in 2014 to 29.45% in 2019. The incidence of AEFI of MPV-AC and MPV-ACYW135 were 49.75 per 100,000 and 45.44 per 100,000, respectively, without statistical difference.

### Conclusion

Children in Hangzhou had high booster immunization of MenV coverage. The use amount and use rate of MPV-ACYW135 increased year by year, indicating more and more parents

**Funding:** Our study was funded by the Hangzhou Science and Technology Development Guide Plan [grant number: 20171226Y25]. The funders had no role in study design, data collection and analysis, decision to publish, or preparation of the manuscript.

**Competing interests:** The authors have declared that no competing interests exist.

had chosen MPV-ACYW135 as an alternative to MPV-AC at their own expense for children. The use proportions of MPV-ACYW135 were different in urban, suburban and rural areas. Both MPV-AC and MPV-ACYW135 were safe for children.

## Introduction

Meningococcal meningitis is a severe acute respiratory infection caused by neisseria meningitidis (Nm) that may lead fevers, rashes, meningeal irritations, meningitis and even deaths [1]. Apart from meningitis and septicaemia, the meningococcal disease occasionally causes arthritis, myocarditis, pericarditis and endophthalmitis [2]. Globally, meningococcal meningitis has become one of the most serious public health problems in many countries, affecting about 1.2 million people annually. Its mortality is up to 40% and about 20% of the cases are associated with long-term sequelae (such as neurological complications) [3]. The disease has a high incidence in Africa, known as the African meningitis belt which was defined by Molesworth and his coworkers in 2002 [4], and a low incidence in Europe and the United States [5]. There are 12 serogroups, divided into groups A, B, C, D, E, H, I, K, L, X, Y, Z and W135, but most invasive meningococcal infections were caused by groups A, B, C, X, Y and W-135, accounting for 95% of the meningococcal cases [6]. In Asia, particularly in developing countries, meningococcal epidemics of serogroups A and C have resulted in high morbidity and mortality, and the threat remains [7, 8]. In China, serogroup A or serogroup C meningococcal meningitis epidemic occurred in several provinces in late 2004 and early 2005 [9]. In recent years, most cases have occurred in children under 15 years of age, especially in infants aged between 6 months and 2 years [10].

Prophylactic medication can help prevent meningococcal meningitis at individual level, such as among family members and the close contacts of meningitis patients. However, its effectiveness of reducing disease risk in population level in response to the outbreaks of meningococcal disease may be limited [11]. Meningococcal vaccination is the most effective way to prevent meningococcal meningitis. In 1980s, group A meningitis vaccine had been widely used in China, and the incidence of meningococcal meningitis decreased significantly. Since 2006, group A meningococcal polysaccharide vaccine (MPV-A) and MPV-AC began to be administered free to children as EPI vaccines in Hangzhou, China. Children aged 6–18 months were qualified to receive two doses of MPV-A in at least three months apart, followed by two doses of MPV-AC booster at age 3 and 6 years. MPV-ACYW135 entered the market in Hangzhou in 2014 as a self-funded vaccine, mainly replacing MPV-AC for the children at the age of 3 or 6 years old. MPV-ACYW135 has not been included in the free EPI vaccination for the time being because the national financial resources cannot afford the cost of MPV-ACYW135 and the vaccine production capacity cannot currently meet the needs of all children.

The objective of our study is to understand the coverage rates of booster immunization of meningococcal meningitis vaccine, the use rate of MPV-AC and the trend in the use amount of MPV-ACYW135, in Hangzhou between 2014 and 2019.

## Methods

### Setting

Hangzhou is a metropolis in Zhejiang province of East China, with more than 10 million population. There are fifteen districts in Hangzhou, six of which are classified as urban areas

(Shangcheng, Xiacheng, Jianggan, Gongshu, Xihu, and Xihufengjingmingsheng); six of which are suburb area (Binjiang, Xiaoshan, Yuhang, Qiantang, Fuyang, Linan); and the others are rural areas (Tonglu, Jiande, Chunan). Hangzhou has 200 vaccination clinics responsible for vaccinating all the children and adults living in the city, no matter they are local residents or migrants. EPI clinicians must inform parents the benefits and risks of MPV-AC or MPV-A-CYW135 before vaccination.

Since 2014, China began to use the newly developed CIMSFIP to collect routine immunization report data. The routine immunization coverage of vaccines in the National Immunization Program (NIP) in China including Hangzhou were reported and summarized through this system. Our research team derived and analyzed the data from NIP in October 2020.

AEFI refers to the reaction during or after vaccination that may cause damage to tissues, organs, or functions of the recipients that are suspected to be related to vaccination. At present, Chinese national adverse event following immunization information system (CNAEFIIS) is the only designated system for collecting AEFI in China. According to the causes, the adverse events can be divided into the adverse reactions (including common adverse reactions and rare adverse reactions), the vaccine quality accidents, the implementation errors, coincidences and the psychogenic reactions. Our research team derived and analyzed data from CNAEFIIS in October 2020.

## Vaccination status and AEFI collection

According to the NIP vaccine immunization procedures, the study reports the number of targeted children, the number of children vaccinated and the inoculation rates in the jurisdiction. The content of the report includes information such as vaccine, doses, household registrations and the coverage of vaccine. When a non-NIP vaccine is used as a replacement of NIP vaccination, its number of targeted children and vaccinated children will be included in the NIP vaccine routine immunization report. NIP data were derived from the "SAAS Vaccination System". When duplicate information appeared, we matched the child's name, birth date, parents' names and telephone number, combined the vaccination data and reported it. AEFI of MPV-AC and MPV-ACYW135 were exported from CNAEFIIS.

## Statistical analysis

Annual incidences of AEFI in MPV-AC and MPV-ACYW135 were analyzed statistically. MenV AEFI per 100,000 populations was calculated using the number of MenV AEFI cases divided by the number of MenV vaccinated. All statistical analyses and graphs were made by SPSS statistical software for Windows (version 17.0, SPSS Inc., Chicago, IL, USA). A value of $P<0.05$ (2-sided) was considered statistically significant. The chi-square test was used to compare the proportions of children who used MPV-AC and MPV-ACYW135, the incidence of common adverse reaction and rare adverse reaction in MPV-AC and MPV-ACYW135. The chi-square trend test was also used to compare the proportion of children using MPV-AC or MPV-ACYW135 year by year.

## Ethical considerations

This study was determined to be exempt from ethical review by the Hangzhou CDC institutional review board. The extraction of data from NIP and CNAEFIIS was safe, which was not linked to individual identifiers.

**Table 1. Reported immunization coverage rates of MenV by dose from 2014 to 2019 in Hangzhou.**

| year | MPV-3[a] | | | MPV-6[b] | | |
|---|---|---|---|---|---|---|
| | No. of children | No. of vaccination | coverage rates (%) | No. of children | No. of vaccination | coverage rates (%) |
| 2014 | 124466 | 121906 | 97.94 | 83953 | 81790 | 97.42 |
| 2015 | 131473 | 131190 | 99.78 | 86282 | 86041 | 99.72 |
| 2016 | 118584 | 118350 | 99.80 | 93570 | 93328 | 99.74 |
| 2017 | 142727 | 142427 | 99.79 | 108336 | 108067 | 99.75 |
| 2018 | 116239 | 116026 | 99.82 | 116551 | 116281 | 99.77 |
| 2019 | 149305 | 148978 | 99.78 | 112746 | 112535 | 99.81 |
| Total | 782794 | 778877 | 99.50 | 601438 | 598042 | 99.44 |

[a]: meningococcal polysaccharide booster vaccine given to children aged 3 years

[b]: meningococcal polysaccharide booster vaccine given to children aged 6 years

## Results

### MenV coverage

1,376,919 infants from 2014 to 2019 were registered in HZIIS. There were 124,466, 131,473, 118,584, 142,727, 116,239 and 149,305 children each year. The coverage rates were 99.50% in MenV-3 and 99.44% in MPV-6, consistently over 95% from 2014 to 2019 (Table 1).

### Trend of MenV vaccination

From 2014 to 2019, the proportion of children using MPV-AC decreased year by year, from 87.37% in 2014 to 70.55% in 2019 ($\chi^2$ = 18886.42, $P_{\text{for trend}}$<0.05). The proportion of children using MPV-ACYW135 increased year by year, from 12.63% in 2014 to 29.45% in 2019 ($\chi^2$ = 18886.42, $P_{\text{for trend}}$<0.05) (Table 2).

The proportion of children who used MPV-ACYW135 was highest in urban (46.88%), and lowest in suburb (5.87%) ($\chi^2_{\text{urban, suburb}}$ = 292375.33, $\chi^2_{\text{urban, rural}}$ = 15064.35, $\chi^2_{\text{suburb, rural}}$ = 68993.42, all $P$-value<0.05) (Table 3).

### AEFI of MPV-AC and MPV-ACYW135

For 2014 to 2019, 535 AEFI cases of MPV-AC and 137 AEFI cases of MPV-ACYW135 were reported in CNAEFIIS. The incidences of AEFI in MPV-AC and MPV-ACYW135 were 49.75 per 100,000 and 45.44 per 100,000($\chi^2$ = 0.89, $P$>0.05), respectively. The incidences of common adverse reaction in MPV-AC and MPV-ACYW135, mainly the injection site erythema, injection site pain and fever, were 39.98 per 100,000 and 31.18 per 100,000, respectively ($\chi^2$ = 4.80,

**Table 2. The proportion of children using MPV-AC or MPV-ACYW135.**

| year | No. Vaccinated | MPV-AC | | MPV-ACYW135 | |
|---|---|---|---|---|---|
| | | No. | Proportion (%) | No. | Proportion (%) |
| 2014 | 203696 | 177975 | 87.37 | 25721 | 12.63 |
| 2015 | 217231 | 171927 | 79.14 | 45304 | 20.86 |
| 2016 | 211678 | 169188 | 79.93 | 42490 | 20.07 |
| 2017 | 250494 | 199689 | 79.72 | 50805 | 20.28 |
| 2018 | 232307 | 172176 | 74.12 | 60131 | 25.88 |
| 2019 | 261513 | 184495 | 70.55 | 77018 | 29.45 |
| Total | 1376919 | 1075450 | 78.11 | 301469 | 21.89 |

**Table 3. The proportion of children using MPV-ACYW135 by geography.**

| year | | urban | | | suburb | | | rural | |
|---|---|---|---|---|---|---|---|---|---|
| | No. Vaccinated | No. ACYW135 | Proportion (%) | No. Vaccinated | No. ACYW135 | Proportion (%) | No. Vaccinated | No. ACYW135 | Proportion (%) |
| 2014 | 70823 | 19661 | 27.76 | 111520 | 2129 | 1.91 | 21353 | 3931 | 18.41 |
| 2015 | 75735 | 33674 | 44.46 | 119588 | 4204 | 3.52 | 21908 | 7426 | 33.90 |
| 2016 | 72054 | 30944 | 42.95 | 118317 | 5162 | 4.36 | 21307 | 6384 | 29.96 |
| 2017 | 85280 | 39839 | 46.72 | 141686 | 4775 | 3.37 | 23528 | 6191 | 26.31 |
| 2018 | 76284 | 42996 | 56.36 | 132819 | 10327 | 7.78 | 23204 | 6808 | 29.34 |
| 2019 | 83869 | 50420 | 60.12 | 152804 | 18989 | 12.43 | 24840 | 7609 | 30.63 |
| total | 464045 | 217534 | 46.88 | 776734 | 45586 | 5.87 | 136140 | 38349 | 28.17 |

**Table 4. AEFI incidence of MPV-AC and MPV-ACYW135 vaccinated from 2014 to 2019.**

| Vaccine | | Common adverse reaction | | Rare adverse reaction | | Coincidental event | | Total | |
|---|---|---|---|---|---|---|---|---|---|
| | No. Vaccinated | No. of cases | Reporting rate (/100 000 doses) | No. of cases | Reporting rate (/100 000 doses) | No. of cases | Reporting rate (/100 000 doses) | No. of cases | Reporting rate (/100 000 doses) |
| MPV-AC | 1075450 | 430 | 39.98 | 88 | 8.18 | 17 | 1.58 | 535 | 49.75 |
| MPV-ACYW135 | 301469 | 94 | 31.18 | 41 | 13.60 | 2 | 0.66 | 137 | 45.44 |

$P<0.05$). The incidence of rare adverse reaction in MPV-AC (8.18 per 100,000) was lower than that of MPV-ACYW135 (13.60 per 100,000) ($\chi^2 = 7.38$, $P<0.05$) (Table 4). Allergic rash was the main rare adverse reaction for both vaccines, and few cases of epilepsy, febrile convulsion, and angioedema were reported. There were no serious reactions or deaths.

## Discussion

Our study showed that children in Hangzhou had a high coverage of MenV booster vaccination. The proportions of MPV-AC or MPV-ACYW135 use changed by year, and MPV-ACYW135 was used more and more frequently for children from 2014 to 2019. Our finding demonstrated a positive safety profile of MenV with reasonable low incidences of AEFI for both MPV-AC and MPV-ACYW135. Urban children seem to be more likely use MPV-ACYW135 than suburb's.

MenV has been continuously used for many years in Hangzhou. During 2014 to 2019, the coverage rates of the third or fourth dose of MenV were all above 95%, which was similar to the reported coverage rates of MenV in NIP in China 2014 (98.54% and 98.53%) [12], and was higher than the one in Dong Weibo's research which was lower than 95% in Fenghua, Zhejiang [13], and the one in Xie Qun's research in Xiamen, Fujian [14]. With the high immunization level of MenV, the incidence of meningococcal meningitis decreased. From 2006 to 2017, the morbidity and mortality of meningococcal meningitis in Zhejiang province decreased gradually by year. Since 2010, less than 10 cases a year have been reported in Zhejiang [15], especially in Hangzhou (S1 Table), meaning MenV effectively prevented the occurrence of meningococcal meningitis. In recent years, there were few cases of meningococcal meningitis in Hangzhou.

Based on the researches in recent years, MPV-AC had good immunogenicity in children aged between 2–6 years. It can achieve good protective effect one month after immunization, and its effect decrease two years after immunization [16]. A booster immunization with MPV-AC is necessary (S2 Table), which induces good immune response after primary immunization with either MPV-A or MCV-AC [17]. In the 20th century, there have been changes in epidemic flora of Nm in different countries and regions and the need for MPV-ACYW135

increase. After 2006, there have been reports of sporadic cases of group W135 [18, 19]. The surveys of Nm carriers in some areas of China show that healthy people's serum bactericidal antibodies of group Y and W135remain low, which indicates that there is a risk of epidemic related flora [20–22].

The incidence of AEFI in MPV-AC and MPV-ACYW135 was lower in Hangzhou than Shanghai [23] and higher than Fujian [24]. There was no difference in the incidence of AEFI between MPV-AC and MPV-ACYW135, with allergic rash and urticaria which were the main events, and no more serious damage was caused, indicating that both vaccines were safe.

The proportion of children who used MPV-ACYW135 was highest in urban and lowest in suburb. The suburbs had the highest total number of MenV vaccinated, but the lowest rate of MPV-ACYW135. The economic income in rural had always been at a low level (S3 Table), but the vaccination rate of MPV-ACYW135 was not the lowest. Therefore, economic reasons may not be the main reason for parents to choose MPV-ACYW135 at their own expense.

In sum, the MPV-ACYW135 has stable clinical performance [25] and good immunological effects [26, 27], which are relatively safe for children. In particular, in order to make MPV-ACYW135 available to all children regardless of geographical location and promote health equity, the government should consider updating the current meningococcal meningitis immunization strategy in China to use MPV-ACYW135 for booster immunization in children at 3 years and 6 years old.

### Limitations of the study

Due to the new CNAEFIIS system permissions and data conversion problems, AEFI data cannot be sorted by district or county for the time being, and it is impossible to compare AEFI reporting rates between regions.

A second limitation of our study was the findings that the suburb had the largest population and inoculation amount in Hangzhou, but its inoculation proportion of MPV-ACYW135 was not high. What causes the low inoculation proportion of MPV-ACYW135 in the suburb is a problem that needs to be further studied.

### Conclusions

To sum up, children in Hangzhou had high booster immunization with MenV. In the case that MPV-AC and MPV-ACYW135 were equally safe, more and more children's parents chose MPV-ACYW135 at their own expense instead of MPV-AC. The use proportions of MPV-ACYW135 were different in urban, suburban and rural areas.

### Supporting information

**S1 Table. Month distribution of meningococcal meningitis in Hangzhou from 1950 to 2019(No. of cases).** Since the 1980s, the number and incidence of meningitis in Hangzhou have been gradually decreasing. The periodic prevalence of meningitis has disappeared in Hangzhou and has been sporadic for many years, with zero incidence in some years. In recent years, both urban and suburban areas are in a sporadic state, and there is no cluster epidemic. (DOCX)

**S2 Table. Nm[a] antibody levels in group A and group C of healthy people in an urban area of Hangzhou in 2018.** Another study in Hangzhou showed that the antibody concentration of group A or group C in the subjects would increase with increasing immunization times, and the average serum antibody concentration was the highest in people who received 4 doses of meningococcal vaccine. It can be seen that the two doses of basic immunization within 18

months of age and one dose of booster immunization at 3 and 6 years of age can produce high concentration of antibodies.
(DOCX)

**S3 Table. Hangzhou's per capita income over the years from 2016 to 2018.** Hangzhou's financial per capita income from 2016 to 2018 shows that the urban per capita income is much higher than the rural per capita income. The disposable income of rural people is much lower than that of urban people.
(DOCX)

## Acknowledgments

We thank the staffs at county level Centers for Disease Control and Prevention and in vaccination clinics in Hangzhou for their vaccination service.

## Author Contributions

**Conceptualization:** Xinren Che.

**Data curation:** Xinren Che, Yan Liu.

**Formal analysis:** Yuyang Xu, Xuechao Zhang, Wei Jiang.

**Investigation:** Xinren Che, Jun Wang, Jian Du, Xiaoping Zhang.

**Methodology:** Xinren Che, Wenwen Gu.

**Writing – original draft:** Xinren Che.

**Writing – review & editing:** Xinren Che.

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
