## [Decision Letter · Decision Letter 0]

24 Feb 2021

PONE-D-21-02890

Booster immunization of meningococcal meningitis vaccine among children in Hangzhou, China, 2014-2019

PLOS ONE

Dear Dr. Liu,

Thank you for submitting your manuscript to PLOS ONE. After careful consideration, we feel that it has merit but does not fully meet PLOS ONE’s publication criteria as it currently stands. Therefore, we invite you to submit a revised version of the manuscript that addresses the points raised during the review process.

The expert reviewers appreciate that this is a relevant topic of study.  However, the interpretation made and the basis of recommending one vaccine versus another is unclear based on the data and statistics presented.  It is also unclear whether these findings have a broader relevance beyond the region in which this study was conducted.  I urge you to address these important aspects should you decide to submit a revised version of this manuscript.

We look forward to receiving your revised manuscript.

Kind regards,

Ashlesh K Murthy, M.D., Ph.D.

Academic Editor

PLOS ONE

Journal Requirements:

2. Thank you for providing the date(s) when patient medical information was initially recorded. Please also include the date(s) on which your research team accessed the databases/records to obtain the retrospective data used in your study.

3. In your ethics statement in the Methods section and in the online submission form, please provide additional information about the data used in your retrospective study. Specifically, please confirm whether all data were fully anonymized before you accessed them.

4. We note you have included a table to which you do not refer in the text of your manuscript. Please ensure that you refer to Table 1 in your text; if accepted, production will need this reference to link the reader to the Table.

Reviewers' comments:

Reviewer's Responses to Questions

**Comments to the Author**

1. Is the manuscript technically sound, and do the data support the conclusions?

Reviewer #1: Partly

Reviewer #2: Partly

2. Has the statistical analysis been performed appropriately and rigorously? 

Reviewer #1: Yes

Reviewer #2: Yes

3. Have the authors made all data underlying the findings in their manuscript fully available?

Reviewer #1: Yes

Reviewer #2: Yes

4. Is the manuscript presented in an intelligible fashion and written in standard English?

Reviewer #1: No

Reviewer #2: Yes

5. Review Comments to the Author

Reviewer #1: Here are my comments:

Typo: line 33 China Information Management System For Immunization Programming (CIMSFIP)

Line 66 A large serogroup A and Serogroup C???

Line 74 and 79 two dots

P value should be lower case in line 116

Typo in Table1) title is 2014-2018, should be 2014-2019

Line 123 the number should be 1,376,919 and not 1376,919

Use correct verb for the vaccine (not vaccined/ vaccinated is correct)

Line 158 fourth dose is correct

Line 200 We believe

In lines 105-107 authors mentioned about household registration. Why not having more info about rural and urban vaccination distribution in the study? Which location have received more MPV-ACYW135 and why? Any idea about “more populous municipalities had lower AEFI reporting rate than less populous ones”?

Please describe why the MPV-ACYW135 is self-paid? What are the reasons that government does not support its free or subsidized use of MPV-ACYW135 vaccination?

Any information about duplicate records and how did you handle it? Wu et al. (2019) found that “ an average of 3% duplicate records within provinces[China], and that duplicated record rates were higher in the eastern region than the western region”

Please check https://doi.org/10.1016/j.vaccine.2019.08.070

Reviewer #2: In this study, the authors investigated the coverage rates of booster immunization of meningococcal meningitis vaccines, and the proportion of use of MPV-AC and the change in the number of MPV-ACYW135 used, in Hangzhou between 2014 and 2019. They concluded that the reported immunization coverage rates of booster immunization of MenV were all above 95%, and children in Hangzhou had a high coverage of MenV booster vaccination. The proportion of children using MPV-ACYW135 increased from 12.63% in 2014 to 29.45 in 2019. The incidence rate of AEFI of MPV-AC and MPVACYW135 was 49.75 per 100,000 and 45.44 per 100,000, respectively. However, some conclusions cause concern, which must be addressed.

1. Line 198 The authors claimed, “we believe that MPV-ACYW135 has advantages of simultaneously preventing a variety of diseases, reducing the number of injections and simplifying immunization procedures,” however, the authors did not provide any data such as serum antibody concentration in Hangzhou, cost-effective to demonstrate the MPV-ACYW135 better than MPV-AC. Statistical analysis also show there is no difference in the incidence of AEFI between MPV-AC and MPV-ACYW135. All data only show children in Hangzhou had a high coverage of MenV booster vaccination, the number of MPV-ACYW135 was used more frequently for children from 2014 to 2019. Why authors recommended to use MPV-ACYW135 instead of MPV-AC for free in Chinese children aged 3-6 years?

2. Discussion: line 186-195 cause controversy. For an example, “The AEFI data used in this paper was passively monitoring data, which may not truly reflect the occurrence of AEFI of MPV-AC or MPV195-ACYW135 to some extent.” Need to rewrite this paragraph.

3. Minor error: line 127 from 2014 to 2018 in Hangzhou should be from 2014 to 2019 in Hangzhou.

Line 312 S3 Nma antibody levels May be S2 Nma antibody levels. (there no S3 table).

6. PLOS authors have the option to publish the peer review history of their article (what does this mean?). If published, this will include your full peer review and any attached files.

Reviewer #1: No

Reviewer #2: No

---

## [Author Response · Author response to Decision Letter 0]

31 Mar 2021

Response to the “Journal Requirements”:

1. Please ensure that your manuscript meets PLOS ONE's style requirements, including those for file naming. The PLOS ONE style templates can be found at……

Authors’ response:

According to the requirements, we have checked the manuscript and ensure that our manuscript meets PLOS ONE's style requirements. If there are any formatting errors that we do not recognize, please specify them.

2. Thank you for providing the date(s) when patient medical information was initially recorded. Please also include the date(s) on which your research team accessed the databases/records to obtain the retrospective data used in your study.

Authors’ response:

As suggested, we have added the date(s) on which our research team accessed the databases/records used in our study in lines 102-103 as follows:” Our research team derived and analyzed the data from NIP in October 2020.”

3. In your ethics statement in the Methods section and in the online submission form, please provide additional information about the data used in your retrospective study. Specifically, please confirm whether all data were fully anonymized before you accessed them.

Authors’ response:

As suggested, we have added ethics statement in the Methods section and in lines 134-137 as follows:” Ethical considerations

This study was determined to be exempt from ethical review by the Hangzhou CDC institutional review board. The extraction of data from NIP and CNAEFIIS was safe, which was not linked to individual identiﬁers.”

4. We note you have included a table to which you do not refer in the text of your manuscript. Please ensure that you refer to Table 1 in your text; if accepted, production will need this reference to link the reader to the Table.

Authors’ response:

As suggested, we have linked the reader to the Table in line 143.

Response to the reviewer's comments:

Response to the reviewer #1

1. Typo: line 33 China Information Management System For Immunization Programming (CIMSFIP)

Authors’ response:

As suggested, we have corrected the word ”programming” in line 33.

Line 66 A large serogroup A and Serogroup C???

Authors’ response:

As suggested, we have rephrased sentence in line 66 as follows:” serogroup A or serogroup C.”

Line 74 and 79 two dots

Authors’ response:

As suggested, we have deleted one dot in line 74.

P value should be lower case in line 116

Authors’ response:

As suggested, we have changed “P value” into “P value” in line 128.

Typo in Table1) title is 2014-2018, should be 2014-2019

Authors’ response:

As suggested, we have changed “2014-2018” into “2014-2019” in line144.

Line 123 the number should be 1,376,919 and not 1376,919

Authors’ response:

As suggested, we have changed “1376,919” into “1,376,919” in line 140.

Use correct verb for the vaccine (not vaccined/ vaccinated is correct)

Authors’ response:

As suggested, we have checked all verb for the vaccine in lines 114,117,126,153,157,169,205.

Line 158 fourth dose is correct

Authors’ response:

As suggested, we have used the right “fourth” in line 178.

Line 200 We believe

Authors’ response:

We reedited the entire text of the paragraph where "we believed" is.

In lines 105-107 authors mentioned about household registration. Why not having more info about rural and urban vaccination distribution in the study? Which location have received more MPV-ACYW135 and why? Any idea about “more populous municipalities had lower AEFI reporting rate than less populous ones”?

Authors’ response:

As suggested, we have added analysis about rural and urban vaccination distribution in the study in lines 154-157,204-209. AEFI data cannot be summarized by region due to system upgrade, so this study did not analyze and discuss it.

Please describe why the MPV-ACYW135 is self-paid? What are the reasons that government does not support its free or subsidized use of MPV-ACYW135 vaccination?

Authors’ response:

As suggested, we have explained the reason “why the MPV-ACYW135 is self-paid” in line 81-85.

Any information about duplicate records and how did you handle it? Wu et al. (2019) found that “ an average of 3% duplicate records within provinces[China], and that duplicated record rates were higher in the eastern region than the western region”

Please check https://doi.org/10.1016/j.vaccine.2019.08.070

Authors’ response:

As suggested, we have checked “https://doi.org/10.1016/j.vaccine.2019.08.070” and explain how did we handle duplicate records in 119-122.

Response to the reviewer #2

1. Line 198 The authors claimed, “we believe that MPV-ACYW135 has advantages of simultaneously preventing a variety of diseases, reducing the number of injections and simplifying immunization procedures,” however, the authors did not provide any data such as serum antibody concentration in Hangzhou, cost-effective to demonstrate the MPV-ACYW135 better than MPV-AC. Statistical analysis also show there is no difference in the incidence of AEFI between MPV-AC and MPV-ACYW135. All data only show children in Hangzhou had a high coverage of MenV booster vaccination, the number of MPV-ACYW135 was used more frequently for children from 2014 to 2019. Why authors recommended to use MPV-ACYW135 instead of MPV-AC for free in Chinese children aged 3-6 years?

Authors’ response:

As suggested, we agree with Reviewer 2. We have revised our conclusions in lines 224-228 and made changes to the discussion section.

2. Discussion: line 186-195 cause controversy. For an example, “The AEFI data used in this paper was passively monitoring data, which may not truly reflect the occurrence of AEFI of MPV-AC or MPV195-ACYW135 to some extent.” Need to rewrite this paragraph.

Authors’ response:

As suggested, we have rephrased this paragraph in lines 216-223.

3. Minor error: line 127 from 2014 to 2018 in Hangzhou should be from 2014 to 2019 in Hangzhou.

Line 312 S3 Nma antibody levels May be S2 Nma antibody levels. (there no S3 table).

Authors’ response:

As suggested, we have changed “2014-2018” into “2014-2019” in line144 and “S2 Nma antibody levels”in line 326.

---

## [Decision Letter · Decision Letter 1]

29 Apr 2021

Booster immunization of meningococcal meningitis vaccine among children in Hangzhou, China, 2014-2019

PONE-D-21-02890R1

Dear Dr. Liu,

We’re pleased to inform you that your manuscript has been judged scientifically suitable for publication and will be formally accepted for publication once it meets all outstanding technical requirements.

Kind regards,

Ashlesh K Murthy, M.D., Ph.D.

Academic Editor

PLOS ONE

Additional Editor Comments (optional):

Reviewers' comments:

Reviewer's Responses to Questions

**Comments to the Author**

1. If the authors have adequately addressed your comments raised in a previous round of review and you feel that this manuscript is now acceptable for publication, you may indicate that here to bypass the “Comments to the Author” section, enter your conflict of interest statement in the “Confidential to Editor” section, and submit your "Accept" recommendation.

Reviewer #1: All comments have been addressed

Reviewer #2: All comments have been addressed

2. Is the manuscript technically sound, and do the data support the conclusions?

Reviewer #1: Yes

Reviewer #2: Partly

3. Has the statistical analysis been performed appropriately and rigorously? 

Reviewer #1: Yes

Reviewer #2: Yes

4. Have the authors made all data underlying the findings in their manuscript fully available?

Reviewer #1: Yes

Reviewer #2: Yes

5. Is the manuscript presented in an intelligible fashion and written in standard English?

Reviewer #1: Yes

Reviewer #2: Yes

6. Review Comments to the Author

Reviewer #1: (No Response)

Reviewer #2: In order to demonstrate the MPV-ACYW135 better than MPV-AC ( immunological effects), the authors should compare the serum antibody titer or T cells response after booster immunization of MPV-ACYW135 or MPV-AC.

7. PLOS authors have the option to publish the peer review history of their article (what does this mean?). If published, this will include your full peer review and any attached files.

Reviewer #1: **Yes: **ABOLFAZL GHASEMI

Reviewer #2: No

---

## [Editor Report · Acceptance letter]

14 May 2021

PONE-D-21-02890R1 

Booster immunization of meningococcal meningitis vaccine among children in Hangzhou, China, 2014-2019 

Dear Dr. Liu:

I'm pleased to inform you that your manuscript has been deemed suitable for publication in PLOS ONE. Congratulations! Your manuscript is now with our production department. 

Kind regards, 

on behalf of

Dr Ashlesh K Murthy 

Academic Editor

PLOS ONE